# Viral fusion proteins of classes II and III recognize and reorganize complex biological membranes
Chetan S. Poojari ✉, Tobias Bommer & Jochen S. Hub ✉

Viral infection requires stable binding of viral fusion proteins to host membranes, which contain hundreds of lipid species. The mechanisms by which fusion proteins utilize specific host lipids to drive virus–host membrane fusion remains elusive. We conducted molecular simulations of classes I, II, and III fusion proteins interacting with membranes of diverse lipid compositions. Free energy calculations reveal that class I fusion proteins generally exhibit stronger membrane binding compared to classes II and III — a trend consistent across 74 fusion proteins from 13 viral families as suggested by sequence analysis. Class II fusion proteins utilize a lipid binding pocket formed by fusion protein monomers, stabilizing the initial binding of monomers to the host membrane prior to assembling into fusogenic trimers. In contrast, class III fusion proteins form a lipid binding pocket at the monomer–monomer interface through a unique fusion loop crossover. The distinct lipid binding modes correlate with the differing maturation pathways of classes II and III proteins. Binding affinity was predominantly controlled by cholesterol and gangliosides as well as via local enrichment of polyunsaturated lipids, thereby locally enhancing membrane disorder. Our study reveals energetics and atomic details underlying lipid recognition and reorganization by different viral fusion protein classes, offering insights into their specialized membrane fusion pathways.

The transmission of emerging and re-emerging viruses from animals to humans is a major concern, as outbreaks of viral diseases have devastated humanity and imposed a significant economic burden worldwide. For the majority of viruses, no vaccines are available, or they are only little effective due to adaptability of viruses[1]. Understanding of host–pathogen interactions is vital for the development of therapeutics against viral infections. Many pathogenic viruses are enclosed by a lipid envelope that, upon infection, fuses either with the plasma membrane (PM) or with the endosomal membranes of the host cell[2,3]. Membrane fusion is energetically expensive; therefore, virus surfaces are decorated with fusion proteins that bind to the host membrane and help to overcome the energetic barriers along the fusion pathway[4–11]. Fusion proteins undergo large-scale structural transitions, shifting from a metastable prefusion conformation on the viral surface to an intermediate and stable fusogenic trimer conformation upon interacting with host membranes[12–17]. While recent structural studies have successfully captured the prefusion and postfusion conformations of fusion proteins, our understanding of fusion protein binding to the host membrane remains incomplete.

Viral fusion proteins are grouped into three classes based on structural similarity[1,18]. Class I fusion proteins are characterized by a trimeric α-helical coiled-coil structure in the post-fusion state and are found in respiratory viruses such as influenza viruses, respiratory syncytial virus, and coronaviruses[19–21]. Class II fusion proteins, composed of β-sheet rich domains, are found in arthropod-borne viruses such as Rift Valley fever virus (RVFV), dengue virus, Semliki forest virus, and chikungunya virus[12,22,23]. Class III fusion proteins, adopting a combination of coiled-coil and β-sheet domains, are found in viruses such as pseudorabies virus, herpes simplex viruses, and vesicular stomatitis virus[14–16,24,25].

All viral fusion proteins undergo structural rearrangements before binding to the host membrane and initiating fusion. The structural mechanism leading to the formation of fusogenic trimers varies among fusion protein classes and may be triggered by proteolytic cleavage, change in pH, or binding to host cell protein receptors[26]. Class I proteins are pre-formed homo-trimers in the pre-fusion state and requires proteolytic processing to expose the fusogenic trimeric subunit. Class II proteins exist as pre-fusion homo-dimers or as hetero-dimers with a companion glycoprotein. At low pH, the hetero-dimeric complex dissociates into intermediate monomers, which then bind to the host membrane as monomers before associating into fusogenic trimers[12,13]. Class III proteins are

Theoretical Physics and Center for Biophysics, Saarland University, PharmaScienceHub (PSH), 66123 Saarbrücken, Germany. ✉e-mail: chetan.poojari@uni-saarland.de; jochen.hub@uni-saarland.de

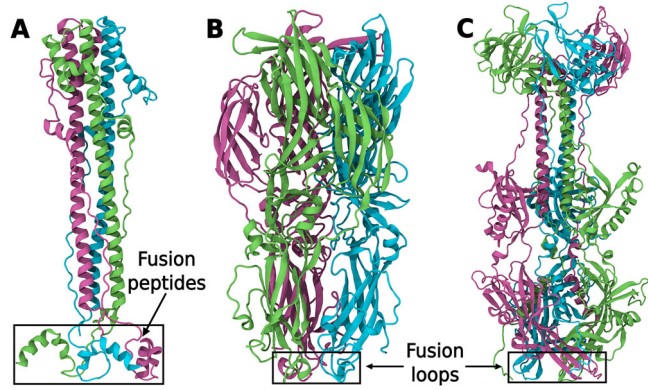

**Fig. 1 | Post-fusion trimer structures of classes I, II and III viral fusion proteins.** Structures of (**A**) influenza hemagglutinin (IAV HA2; PDB IDs: 1QU1[19], 1IBN[69]) (**B**) Rift Valley fever virus Gc protein (PDB ID: 6EGU[22]), and (**C**) pseudorabies virus gB protein (PDB ID: 6ESC[25]) are shown in cartoon representation. The monomers in each protein are colored green, pink and cyan, respectively. Black boxes highlight fusion peptide/loop regions required for binding to the host membrane.

homotrimers in the pre-fusion state, and their structural rearrangement to form fusogenic trimers is triggered by coordinated interactions of multiple glycoproteins[14,17,25]. Across fusion protein classes, fusogenic trimers bound to the host membrane contract, bringing the host and viral membranes into close proximity. Therefore, stable binding at the host membrane is essential to endure the protein–membrane forces required for membrane deformation during fusion. This requires robust and selective protein–lipid interactions and is particularly critical for class II proteins, which assemble from monomers into fusogenic trimers while bound to the host membrane, necessitating stable membrane binding even for monomeric proteins.

The role of lipid membrane composition in binding and fusion has been extensively studied using liposome flotation assays. Previous research highlighted the significance of cholesterol in facilitating initial binding and membrane fusion[22,23,25,27–34]. For instance, the fusion protein of Semliki Forest virus exclusively binds to the membrane in the presence of cholesterol, with fusion loops forming direct contacts with cholesterol, as demonstrated in photo-cholesterol cross-linking experiments[23]. Further, a point mutation (A226V) has been shown to render Chikungunya virus membrane binding dependent on cholesterol[35]. Fusion proteins also recognize phospholipids, as evidenced by the crystal structure of the RVFV protein in complex with short-tailed phosphocholine lipid[22]. Additionally, gangliosides have been implicated in viral entry into host cells[32,36–38]. While these studies highlight the critical role of lipids during viral infection, the structural mechanisms and energetics of host lipid recognition remains unresolved. Moreover, these investigations utilized at most monounsaturated tails (16:0-18:1 or di-18:1). However, recent lipidomic studies of mammalian plasma membrane have unveiled a significantly more complex lipid composition, characterized by variations in tail length and degree of unsaturation[39].

Our recent study further emphasized the importance of lipid tail composition in viral infection by demonstrating that polyunsaturated lipid favor stalk formation during the early stages of membrane fusion[4]. To gain deeper understanding of how viral proteins exploit host lipids during viral entry, it is imperative to consider the extensive repertoire of lipids present in the host membrane. The significance of lipid composition for fusion is further underscored by the observation that viruses manipulate host lipid metabolism during infection, resulting in an increased polyunsaturated and a decreased saturated lipid content[40,41]. However, it remains unclear if and how viruses utilize the enriched polyunsaturated lipids to enhance pathogenicity. This study employs a combination of coarse-grained and atomistic molecular dynamics (MD) simulations, alongside free energy calculations, to quantify and structurally characterize membrane binding of one representative from each fusion protein class. We examined class I influenza hemagglutinin (IAV HA2), class II Gc protein from RVFV, and class III gB

protein from the pseudorabies virus (PrV, Fig. 1). Binding affinities were computed across a range of membrane models, including complex plasma membranes and various binary and ternary lipid mixtures featuring distinct head groups, tail lengths, tail unsaturations, and sterol content. Our simulations reveal a significant impact of lipid composition and protein class on membrane binding, reflecting distinct fusion mechanisms among fusion protein classes. We found that polyunsaturated lipids, when paired with cholesterol greatly enhance binding affinity. Additionally, distinct lipid-binding pockets in classes II and III proteins correlate with their differing maturation pathways, which fusion proteins take en route to forming fusogenic trimers. Notably, mutations in these lipid binding pockets change lipid specificity of the pockets, highlighting their importance during infection. We further demonstrate co-localization of fusion proteins with gangliosides, indicating their role in membrane binding. Our MD simulations align with a rich body of experimental data. This systematic analysis advances our understanding of how viral proteins recognize and reorganize lipids during infection.

## Results

### Membrane binding affinities follow the order class I > class II ≳ class III for many fusion proteins

To investigate the energetics of fusion peptide or loop insertion into a complex membrane, we set up Martini coarse-grained (CG) simulations systems composed of a fusion protein and a plasma membrane (PM) model, with the lipid composition of a typical outer leaflet of a mammalian PM (Table S1). Based on a recent lipidomics study[39], the membrane was composed of 40% cholesterol, 25% phosphatidylcholine (PC), 25% sphingomyelin (SM), 4% glycosphingolipid (GM3), 4% phosphatidylserine (PS), and 2% phosphatidylethanolamine (PE, Table S1). To model the interactions of the viral fusion proteins with host membranes prior to the collapse of the fusion proteins, we simulated the structurally well-characterized post-fusion conformations in the absence of their transmembrane helical anchors. We used umbrella sampling to compute the potential of mean force (PMF) of protein binding to the membrane. According to the PMFs, the binding affinity $\Delta G_{bind}$ to the PM is stronger (more negative) for IAV HA2 ($-115$ kJ/mol) as compared to RVFV Gc ($-82$ kJ/mol) and PrV gB ($-59$ kJ/mol) (Fig. 2 D,E). The stronger membrane anchoring of IAV HA2 is rationalized by the insertion of a large hydrophobic fusion peptide into the hydrophobic core of the membrane, as opposed to RVFV Gc or PrV gB that anchor to the membrane by insertion of only two or one bulky aromatic side chains per monomer, respectively (Figs. 5, 6). In addition, fusion peptides of HA2 are connected to the central helix by flexible loops, which enable the fusion peptides to reorient and insert deep into the membrane, whereas the fusion loops are structurally constrained to the folded protein, thus allowing the insertion of only one or two aromatic side chains.

To determine whether the differences in binding affinities are specific to the simulated HA2, Gc, and gB, or whether they represent general trends among class I, II, and III fusion proteins, we carried out an extensive sequence analysis. We compared sequences of 23 fusion proteins from class I, 33 fusion proteins from class II, and 18 fusion proteins from class III (Fig. 3, Tables S5–S7). Remarkably, among many fusion proteins, the sequence identities among fusion loops/peptides (Fig. 3, upper triangles) exceed by far the sequence identities among the complete fusion proteins (Fig. 3, lower triangles), demonstrating the functional importance of fusion loops/peptides for viral replication.

According to the sequence identity matrices, class I fusion proteins are highly diverse, exhibiting low sequence identity even within the same family. However, similar to IAV HA2, fusion peptides of all class I fusion proteins contain several hydrophobic amino acids, suggesting that many class I fusion proteins may bind to the host membrane with high affinity (Table S5). Class II fusion proteins are highly conserved within the families of hantaviridae, togaviridae and flaviviridae with sequence identities in the range of 40–60% (Fig. 3B, lower triangle), while the respective fusion loops reach sequence identities of more than 80% (Fig. 3B, upper triangle). In agreement with RVFV Gc, many of the class II fusion loops analyzed here

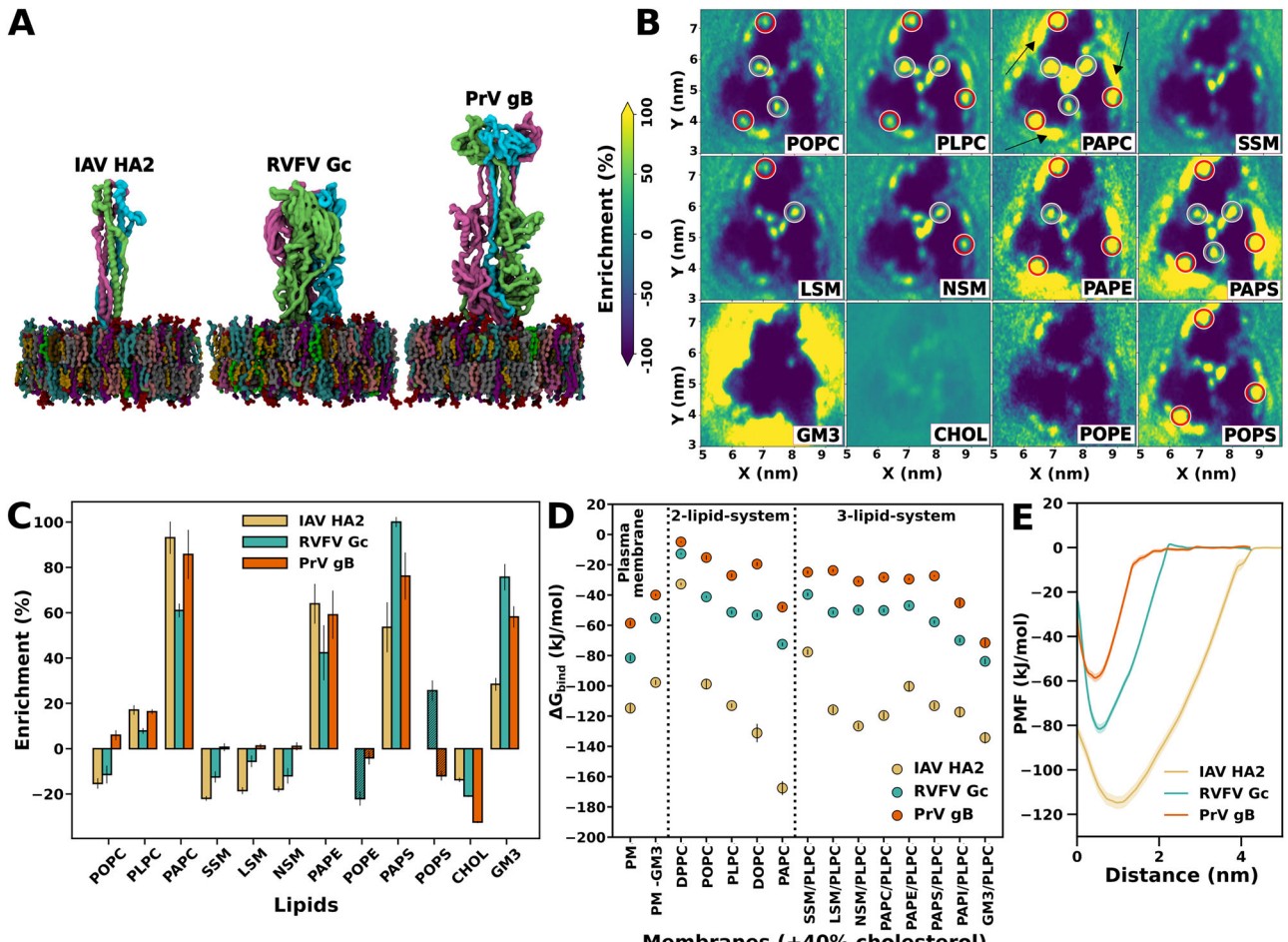

**Fig. 2 | MD simulations of viral fusion proteins bound to lipid membranes.**
**A** Coarse-grained models of the protein-membrane complexes for IAV HA2 (left), RVFV Gc (middle), and PrV gB (right). **B** Two-dimensional lipid density maps for the RVFV Gc bound to the outer leaflet of a plasma membrane model. Lipid types are indicated by labels. Lipid binding sites are marked in red circles (experimentally found binding site[22]) and gray circles (second binding site). Density maps for IAV HA2 and PrV gB are shown in Figs. S2 and S3. **C** Enrichment of lipids in contact with the fusion protein relative to the bulk membrane. For comparison, data for POPE and POPS lipids (shaded boxes) are included from a second set of plasma membrane simulations, where PAPE and PAPS lipids were replaced by POPE and POPS lipids. This comparison demonstrates the importance of polyunsaturated tails for lipid enrichment. Lipid enrichment map for plasma membrane containing POPE and

POPS lipids are shown in the Figs. S4 and S5 (n = 3 simulations). **D** Binding free energies $\Delta G_{bind}$ taken from the free energy minimum of potentials of mean force (PMFs) of fusion protein binding, shown for IAV HA2 (brown), RVFV Gc (blue), and PrV gB (red) binding to membranes with 15 different lipid compositions. Left panel: $\Delta G_{bind}$ for plasma membrane (PM) and for PM depleted by GM3. Middle panel: binary mixtures of 40% cholesterol plus one phospholipid (see labels). Right panel: ternary mixtures of 40% cholesterol, 55% PLPC plus 5% of an additional lipid (see labels). **E** Example PMFs for fusion protein binding, here for binding to the plasma membrane model. The minima and flat regions correspond to the membrane-bound and membrane-unbound states, respectively. Vertical black bars in (**D**) and shaded areas in (**E**) denote 1 SE computed by bootstrapping from n = 54 to n = 86 independent umbrella windows.

contain one tryptophan and one phenylalanine that may insert into the membrane, although several variations exist: fusion loops of togaviridae contain a second phenylalanine, while in fusion loops of certain phenuiviridae tryptophan is replaced with a hydrophobic leucine or isoleucine (Table S6, see underlined Phe/Trp residues). Among class III, fusion proteins exhibit a low degree of conservation with an exception for the baculoviridae family. Notably, albeit low conservation between the complete fusion proteins of baculoviridae and some genera within orthomyxoviridae families, the matrix reveals high conservation between the fusion loops of these families, indicating their evolutionary relation[42] (Fig. 3C, central blocks). Among the class III fusion proteins analyzed here, baculoviridae, orthomyxoviridae, and several herpesviridae contain only one hydrophobic aromatic residue (Phe/Trp) per monomer to insert into the membrane (Table S7). However, variations to this observation exist, as shown by Epstein-Barr virus or several rhabdoviridae containing up to three Phe/Trp residues per monomer.

Thus, although we computed binding affinities in this study only for one representative of each class, the sequence analysis suggests that the

trends in binding affinities obtained for HA2, Gc, and gB likely apply for many other members of the fusion protein classes. Specifically, the strong binding observed for IAV HA2 is expected for many class I fusion proteins, which likewise anchor to the host PM by the insertion of fusion peptides. Since many class II proteins insert two aromatic residues per monomer into the membrane, while most class III proteins insert only one aromatic residue per monomer, we propose that the binding affinities to the PM typically follow class I > class II ≳ class III. Exceptions to this trend exist as suggested by class III proteins with up to three Phe/Trp in their fusion loops, indicating that certain class III proteins may reveal stronger membrane affinity compared to certain class II proteins (Tables S5–S7). Notably, despite the presence of only two or one aromatic sites in the fusion loops of RVFV Gc and PrV gB, we find remarkably strong affinities $\Delta G_{bind}$ of −82 kJ/mol or −59 kJ/mol, respectively, suggesting that the affinity is also based on specific interactions with lipid headgroups. Hence, we quantified and structurally characterized the effects of specific lipids on fusion protein binding, as discussed in the following sections.

## Membrane binding is regulated by polyunsaturated lipids and cholesterol

Viral fusion proteins target the outer leaflet of the host PM, which is enriched with PC lipids of varying tail compositions and cholesterol[39]. PC headgroups and cholesterol play a functional role during viral infection as shown by previous experiments[22,23,25,27–34,43,44]. We studied the effect of lipid type on binding via two types of analysis: (1) by examining lipid enrichment at the protein–membrane interface, and (2) by computing the free energy of fusion protein binding to membranes with various lipid compositions.

Two-dimensional lipid densities revealed that polyunsaturated lipids are enriched at the protein–membrane contact region, suggesting that polyunsaturated lipids favor membrane binding (Figs. 2B, C, S3–S5). For instance, 1-palmitoyl-2-arachidonoyl PC (PAPC, 16:0-20:4) is enriched at the cost of depleted 1-palmitoyl-2-linoleoyl PC (PLPC, 16:0-18:2) or 1-palmitoyl-2-oleoyl PC (POPC, 16:0-18:1) lipids. This trend is maintained upon replacing the PC headgroup with PE or PS headgroups (Fig. 2B, C, S3). Additionally, increasing the tail length or unsaturation of sphingomyelin from C16:0 to C24:0 or C24:1 had only a minor effect lateral distribution of lipids (Fig. 2D and supplementary results). These data reveal that RVFV Gc and PrV gB binding to host membranes generally trigger a local enrichment of polyunsaturated lipids.

To clarify how lipid tails and cholesterol influence $\Delta G_{bind}$, we computed PMFs for fusion protein binding to binary lipid membranes containing 40% cholesterol plus one type of PC lipid with varying tail properties. The PMFs reveal increased binding affinity (more negative $\Delta G_{bind}$), with an increasing number of double bonds in lipid tails (Fig. 2D, center panel). Presence of polyunsaturated PAPC lipid induces increasingly stronger binding for all three fusion proteins, with the largest effect on binding for IAV HA2 (Fig. 2D). To validate the strong binding affinity of IAV HA2, we computed additional PMFs of insertion of a single IAV HA2 fusion peptide into PC or phosphatidylglycerol (PG) membranes (POPC or POPC:POPG 80%:20%), for which experimental binding data are available for comparison. Experiments revealed $\Delta G_{bind}$ of a single IAV HA2 fusion peptide of $-32.35$ kJ/mol or $-31.8$ kJ/mol for POPC or POPC:POPG membranes, respectively[45], in good agreement with our calculated values of $-33.3$ kJ/mol or $-27.9$ kJ/mol, respectively (Fig. S1).

To quantify the effect of cholesterol concentration on membrane binding, we performed PMF calculations using 1,2-dioleoyl PC (DOPC) as a reference lipid, with cholesterol concentration set at 0%, 20%, and 40%. DOPC was selected as the reference lipid because experimental binding data are available for comparison for RVFV Gc and PrV gB[22,25]. In qualitative agreement with experiments, the PMFs demonstrate that RVFV Gc and PrV gB bind to the membrane in a cholesterol-dependent manner (Fig. 4). Notably, these PMFs furthermore reveal a small entry barrier reflecting the energetic cost for moving the hydrophobic fusion loops across the polar headgroup region (Fig. 4A, distance ~1.3 nm). This barrier is reduced at higher cholesterol content, rationalized by the fact that cholesterol acts as spacer between phospholipids, reducing the head group packing density.

In binding experiments, RVFV Gc binding to liposomes required at least 20% cholesterol, and binding increased with cholesterol concentration[22], in line with the trends in our simulations. For PrV gB, binding was reported at 40% cholesterol but not at 0% or 20% cholesterol concentration[25]. These data are likewise in qualitative agreement with our simulations as we find generally weaker binding of gB as compared to Gc, rationalizing the requirement for an increased cholesterol content for obtaining stable binding.

The dependence of cholesterol during fusion and infection has been previously demonstrated in other members of classes II and III[23,43,44], but exceptions do exist, as dengue virus and yellow fever virus, do not rely on cholesterol for their membrane-binding process[23]. IAV HA2 binds to the membrane with a strong affinity of $-80$ kJ/mol independent of the presence of cholesterol. This result aligns with biochemical experiments showing that the influenza virus can interact with liposomes even without cholesterol[46].

Although the trends in binding affinities estimated from our simulations align with the experimental findings, the structural mechanisms by

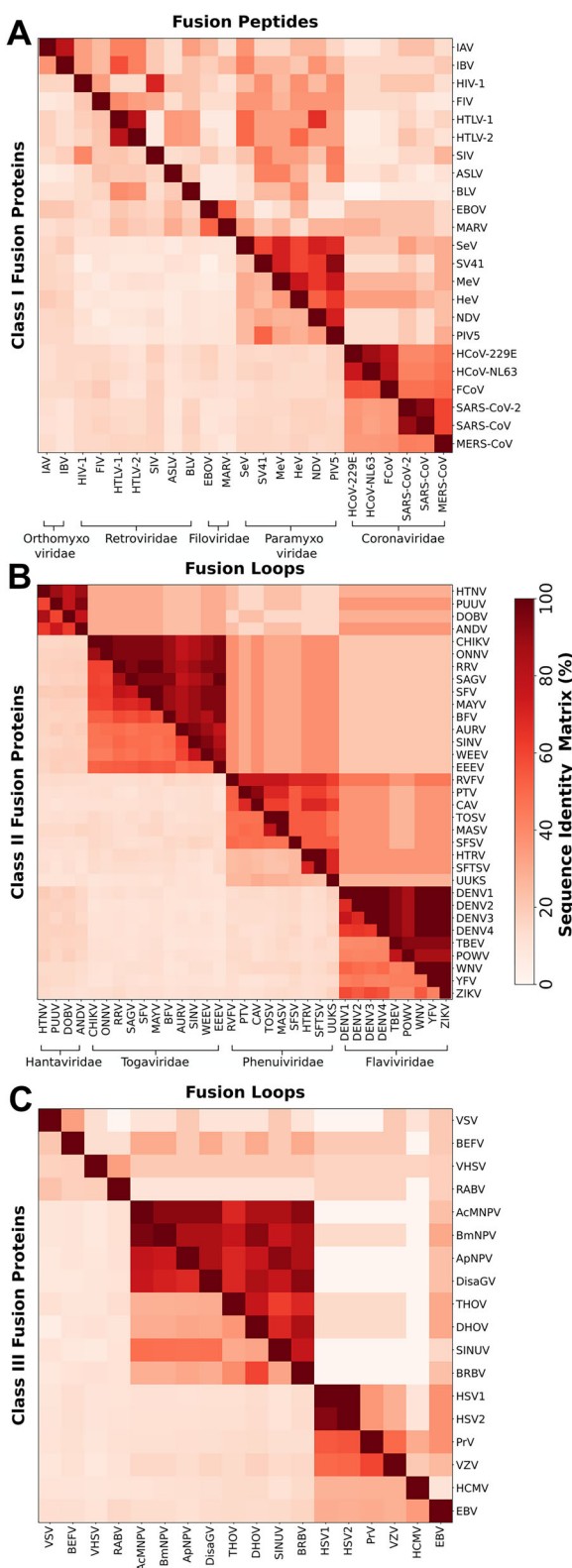

**Fig. 3 | Sequence identity matrices of viral fusion proteins.** Sequence identities are shown for class I (panel **A**), class II (panel **B**), and class III proteins (panel **C**). Lower triangle shows sequence identities for fusion proteins, whereas, upper triangle shows sequence identities for fusion peptides/loops only. See Tables S5–S7 for the full name of viruses, fusion proteins, fusion peptide/loop sequences, and UniProtKB[94] accession number.

which polyunsaturated lipids and cholesterol enhances the binding of RVFV Gc and PrV gB remain to be clarified. Our simulations, together with earlier studies[22,23], rationalize a structural mechanism for enhanced binding of RVFV Gc and PrV gB as follows: (i) cholesterol acts as a spacer between phospholipids, thereby allowing bulky aromatic fusion loop residues to penetrate the head group region during initial binding (Figs. 5C, 6C), while (ii) direct interactions between cholesterol and fusion loop bulky residues may further stabilize the protein–membrane interactions (Figs. S6, S7). (iii) These effects by cholesterol are complemented by enrichment of polyunsaturated lipids that bind to specific lipid-binding sites on the protein

(Figs. 2B, S3–S5), while simultaneously filling the voids beneath the protein with their flexible acyl chains (Figs. 5B, D, E; 6B, D, E; S8; S9). Such dual function would not be possible by lipids with saturated or shorter tails.

Given that polyunsaturated lipids lower the free energy cost for stalk formation, as observed previously[4], we propose that the enrichment of these lipids renders the membrane locally more fusogenic, thereby promoting viral infection (Fig. 2C). Together, the lipid enrichment analyses and binding free energies demonstrate that the binding of viral fusion proteins to membranes is modulated by the combined effects of cholesterol and polyunsaturated lipids.

## Monomer-specific lipid binding pocket in Gc: role in intermediate state and membrane binding

Recent structural studies using X-ray crystallography, cryo-EM, and cryo-ET on class II fusion proteins–such as RVFV Gc (phlebovirus) and E1 from chikungunya and sindbis viruses (alphaviruses) have revealed conserved structural rearrangements by which monomers first bind to the host membrane and, henceforth, assemble to trimers[12,13,47]. Indeed, cryo-EM studies of Sindbis virus[47] revealed bridge-like densities connecting the viral and host membranes, which have been attributed to fusion protein monomers anchored to the host membrane. The varying length of these densities of 15 to 25 nm suggest that monomer are flexible and that the insertion of fusion loops alone may not be sufficient to trigger the assembly into fusogenic trimers. We aim to investigate the potential role of host lipids in stabilizing fusogenic trimers at the host membrane.

Using coarse-grained simulations, we analyzed lipid densities around the RVFV Gc protein. The lipid density maps revealed two distinct types of lipid interactions at the protein surface: (i) highly localized, specific interactions with binding pockets on each monomer (Fig. 2B, red circles), which aligns with a lipid binding pocket revealed by crystallography; (ii) a pocket at each of the three monomer–monomer interfaces (Figs. 2B, grey circles); and (iii) less specific interactions across the protein surface (Fig. 2B, black arrows). To validate the lipid-binding pockets observed in coarse-grained simulations, we back-mapped coarse-grained protein-membrane structures

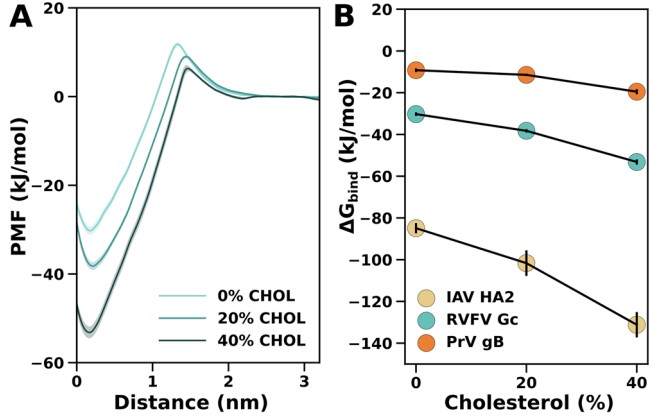

**Fig. 4 | Cholesterol-dependent membrane binding of viral fusion proteins. A** PMFs for RVFV Gc binding to membranes of DOPC plus 0%, 20%, or 40% cholesterol. **B** Binding free energies $\Delta G_{bind}$ taken from PMF calculations for three fusion proteins (see legend) with varying cholesterol concentrations. Shaded areas in (**A**) and vertical black bars in (**B**) denote 1 SE computed by bootstrapping from $n = 54$ to $n = 82$ independent umbrella windows.

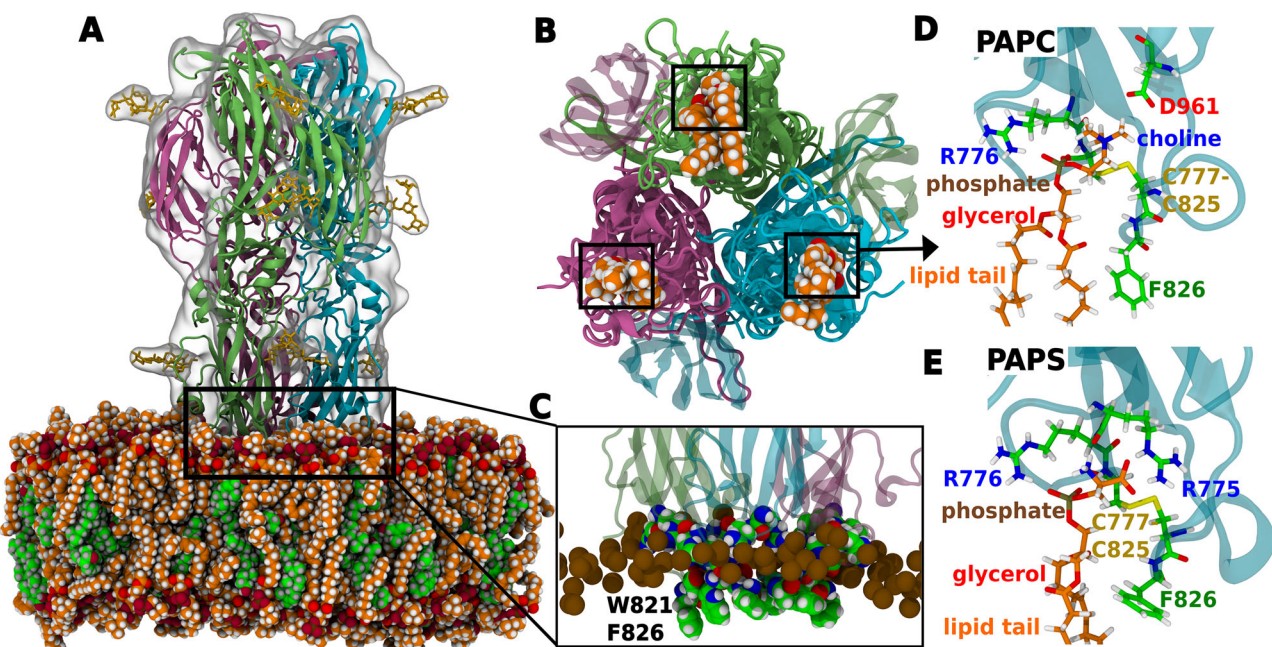

**Fig. 5 | RVFV Gc binds lipids with a pocket formed by each monomer. A** All-atom simulation system of RVFV Gc in contact with a membrane of PAPC (orange/red spheres) and cholesterol (green/red spheres). Gc monomers are shown in green, pink, and cyan cartoon representation, respectively. N-Glycans on the protein are rendered as yellow sticks. Water is not shown for clarity. **B** Bottom view showing a PAPC lipid bound to each Gc monomer (indicated in the black box). **C** Magnified view showing interaction of fusion loops with the membrane. Fusion loop tip

residues W821 and F826 from each monomer insert deep into the apolar region of the membrane below the phosphate atoms (brown spheres), whereas the rim residues R775, R776, H778, H784, N827, and N829 (colored spheres) insert into the interfacial polar region of the membrane. **D, E** PAPC and PAPS lipids in the same promiscuous Gc binding pocket, illustrating interactions with nearby amino acids, respectively. Choline, phosphate, glycerol, and lipid tail are labeled. Lipid tails are shortened for clarity.

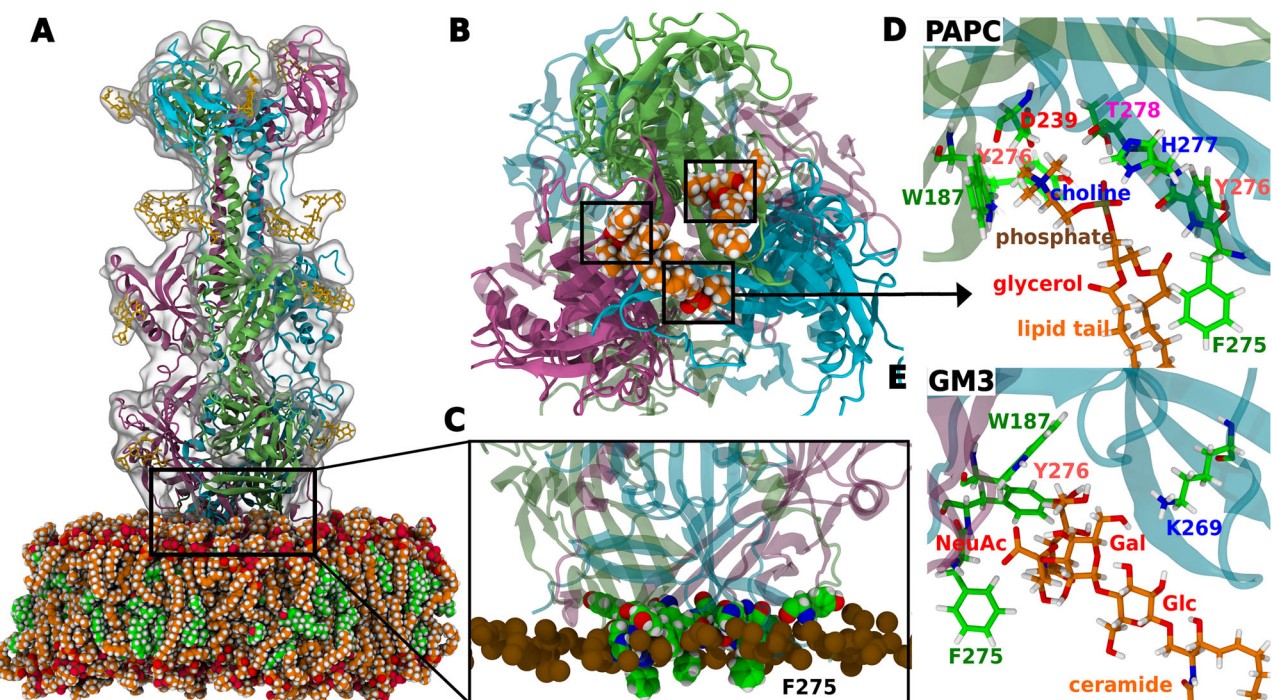

**Fig. 6 | PrV gB bind lipids with a pocket formed at the monomer–monomer interface. A** All-atom simulation system of PrV gB in contact with a PAPC/cholesterol membrane. **B** Bottom view showing PAPC lipid bound at the monomer-monomer interface (indicated in the black box). PAPC is rendered as orange spheres. **C** Magnified view showing interactions of fusion loops with the membrane. Fusion loop tip residue F275 from each monomer inserts into the apolar region of the membrane below the phosphate atoms, whereas the rim residues Y192, W187, Y276 insert into the interfacial polar region of the membrane. **D, E** PAPC and GM3 lipid binding sites and their interactions with nearby amino acids, respectively. Choline, phosphate, glycerol, and lipid tails are labeled for PAPC and headgroup sugars N-acetylneuraminic acid (NeuAc), galactose (Gal), glucose (Glc), and ceramide tail are labeled for GM3. Lipid tails are shortened for clarity. All representations and colors are chosen as in Fig. 5.

to atomistic resolution and conducted additional simulations over several microseconds (Table S4). This approach provided atomistic insights into the binding mechanisms of RVFV Gc to the membrane (Fig. 5).

Host membrane binding occurs through the insertion of bulky fusion loop residues, W821 and F826 from each monomer into the hydrophobic core of the membrane. Additionally, interfacial rim residues (R775, R776, H778, H784, N827, and N829) interact with the polar headgroups of lipids (Figs. 5C). Beyond these non-specific interactions, lipids localize within specific pockets formed by each monomer, consistent with the highly localized densities observed in coarse-grained simulations (Fig. 2B (red circles), Figs. 5, S8). These lipid-binding pockets align with crystallographic data, illustrating how specific lipids fit into binding sites on the protein[22]. For example, the headgroup of polyunsaturated PAPC fits deeply into the binding site, with its choline and phosphate moieties forming hydrogen bonds with the side chains of D271 and R776, respectively, while the polyunsaturated tail (20:4) may pack against the membrane-inserted F826 side chain (Figs. 5D, S8). Interestingly, the same binding site may also accommodate anionic PAPS, as the amino and carboxyl groups of PS may form favorable hydrogen bonds with the backbone oxygen and side chain of R775, respectively (Fig. 5E). The physiological relevance of binding to PS lipids aligns with Gc entry via the endocytic pathway, where endosomal membranes are rich in PS and other anionic lipids. Furthermore, the remarkable promiscuity of the RVFV Gc lipid-binding site offers a structural explanation for the experimentally observed recovery of binding by the Gc D961K mutant upon addition of PS lipids[22]. Indeed, our PMFs show that the D961K mutant exhibits binding affinity similar to the wild-type protein when PS lipids are present in the membrane (Fig. S10).

Based on the presence of distinct lipid-binding pockets, we propose that the experimentally observed metastable flexible monomers[12,13,47] are stabilized on the host membrane by binding specific lipids at monomer-specific binding pockets, subsequently assembling into fusogenic trimers on the membrane.

## Lipid binding at the monomer–monomer interface in gB: implications for fusion mechanisms

The fusion pathway of class III gB proteins involves coordinated interactions among multiple glycoproteins[14,17,25]. However, the precise molecular mechanisms that trigger the conformational change to a fusogenic trimer remain unclear. Additionally, it is unknown whether gB proteins depend on host lipids during fusion. Structural comparisons of pre- and post-fusion trimers suggest that the pre-fusion structure undergoes rearrangements around the central domain, reorienting the fusion loops from facing the viral membrane toward the host membrane[16]. Here, our aim is to elucidate the molecular interactions between the gB protein and the host membrane, specifically investigating whether gB can bind and recognize lipids in a manner similar to the class II Gc protein.

We applied the same simulation protocol used for Gc to analyze lipid distribution around gB. Similar to findings for Gc discussed above, coarse-grained simulations revealed two distinct types of interactions with the gB surface: (i) highly localized, specific interactions with binding pockets formed at the monomer–monomer interface (Figs. S3, S5, red circles) and (ii) broader, less specific interactions across the protein surface (Fig. S3, black arrows). To validate the lipid binding pocket, we again back-mapped the coarse-grained structures to atomistic resolution and performed additional simulations over several microseconds (Table S4), which provided atomistic insights into the membrane binding mechanisms of PrV gB (Fig. 6).

The atomistic simulations showed that gB binds to the membrane through the insertion of the bulky fusion loop residue F275 from each monomer into the membrane's hydrophobic core (Fig. 6C), and via interactions by interfacial rim residues Y192, W187, and Y276 with lipid polar head groups. After the initial contacts, lipids become localized within pockets formed at the monomer–monomer interface (Fig. 6 B, D, E). For instance, the headgroup of polyunsaturated PAPC binds to a groove formed by fusion loops (FL1 and FL2) of two different monomers (Fig. 6D). The

choline headgroup penetrates deeply into this groove, forming hydrogen bonds with residues D239 and Y276 (FL2) and a cation-$\pi$ interaction with W187 in FL1 of monomer 1. Stabilization of PAPC binding is further enhanced by interactions between the lipid phosphate oxygen atoms and both the H277 side chain and the backbone nitrogen of T278 in FL2 of monomer 2. Additionally, a carbonyl oxygen from the lipid ester group forms a hydrogen bond with Y276 in the same loop. Similar to RVFV Gc, the lipid tail packs against the membrane-inserted F275 side chain, contributing to the stability of the interaction.

The structural significance of the lipid binding pocket is revealed by comparing the fusion loop arrangements in pre- and post-fusion states (Fig. S11). In the pre-fusion state, fusion loops from each monomer are spaced widely apart, resembling a tripod-like structure[16,48,49]. However, during the transition to the post-fusion state, these loops move closer and cross over, creating lipid binding pockets at the monomer–monomer interfaces, which promote lipid localization in this region.

### RVFV Gc and PrV gB fusion proteins colocalize with gangliosides on the host membrane

Many viruses utilize specific host protein receptors, yet gangliosides also serve as lipid-based attachment factors in viral entry[14,32,36–38,50,51]. For example, influenza virus fusion protein HA2 binds to ganglioside sialic acid moieties[37,38]. However, it remains unclear whether the fusion proteins Gc and gB interact with gangliosides.

To explore the role of GM3's in viral fusion protein binding, we compared binding free energies with and without GM3. The absence of GM3 led to a ~15–30 kJ/mol reduction in binding affinities across the three fusion proteins studied here (Fig 2D). The lateral density of GM3 around the proteins revealed a nonspecific, dispersed distribution (Fig 2B,C, Figs. S2–S3). Specifically, PrV gB showed GM3 enrichment at the lipid-binding site, that is also accessible to PC lipids (Fig. S3, red circles).

To validate this GM3 binding site, we backmapped coarse-grained PrV gB-PLPC/cholesterol/GM3 system (structure at 15 $\mu$s) to atomistic resolution. During a subsequent two-microsecond atomistic simulation, GM3 remained stably associated with the lipid-binding site, supporting the results observed in CG models (Fig. 6E). This observation aligns with experimental data showing PrV co-localizes with GM1 gangliosides[32]. For RVFV Gc, GM3 exhibited nonspecific binding, though experimental verification is needed. Notably, GM1 has been observed to co-localize with dengue virus, another class II fusion protein like RVFV Gc[36]. Thus, while experiments have established the roles of gangliosides in viral infection[32,36], our study provides structural and energetic insights into gangliosides involvement in viral membrane binding.

### Discussion

The interactions between viral fusion proteins and host membranes are essential for membrane fusion, as robust binding stabilizes the fusion proteins on the host membrane, allowing them to withstand the deformation forces involved in this energetically demanding process. While previous biochemical studies have shown that fusion protein binding is dependent on membrane composition, the structural basis and energetic consequences of these interactions have remained unclear[22,23,25,27–34,36–38,52]. In this study, we examined fusion proteins from classes I, II, and III to show how their binding affinity is shaped by lipid composition and specific protein motifs, which in turn reflect the distinct maturation pathways of each class.

We computed the free energy of membrane binding for three viral fusion proteins using coarse-grained simulations, and we validated the simulations against a wide body of experimental data: the simulations (i) identified the lipid binding site of RVFV Gc in line with crystallographic data[22], (ii) revealed increasing binding affinity with increasing cholesterol content[22,25], (iii) align with effects of the D961K mutant and PS lipids on RVFV Gc binding[22], and (iv) furthermore, binding affinities of the IAV HA2 fusion peptide are in good agreement with experimental data[45]. Thus, the coarse-grained simulations likely provide a reasonable model for estimating binding affinities of viral fusion proteins.

The PMFs reveal that class I fusion protein IAV HA2 binds to all simulated membranes with stronger affinity ($\Delta G_{bind}$ more negative) than class II:RVFV Gc or class III:PrV gB. Sequence conservation analysis across 74 fusion proteins from 13 viral families, focusing on fusion peptides/loops, indicates that these binding affinities are likely representative for many viral fusion proteins and follow the trend: class I > class II $\gtrsim$ class III (Fig. 3, Table S5–S7). The strong plasma membrane binding affinity of a single IAV HA2 trimer ($\Delta G_{bind} = -115$ kJ/mol, Fig. 2D, E) has implications for protein-mediated membrane fusion. Aeffner et al. estimated that the free energy required to form a stalk-like structure in the absence of proteins ranges between ~250 kJ/mol and ~500 kJ/mol, depending on the lipid composition[9]. Thus, binding of at least three IAV HA2 trimers to the plasma membrane would be required to avoid that the membrane-bound HA2 trimers dissociate from the membrane during HA2 collapse, consistent with experimental findings showing that a minimum of three trimers is required to mediate fusion[53–55].

The binding of viral fusion proteins to the host membrane is strongly influenced by lipid composition. Cholesterol promotes binding, despite not being enriched at the protein–membrane interface (Figs. 2B, C, 4). Its role in binding is more intricate: cholesterol acts as a spacer between phospholipids, facilitating the insertion of bulky fusion loop residues into the membrane. Furthermore, we observe direct interactions between cholesterol and fusion loop residues, which align with experimental data (Figs. S6, S7;[23]).

Additionally, the binding affinity is enhanced by increased lipid tail unsaturation. Polyunsaturated lipids impose greater tail disorder, which is compatible with the membrane pertubations required for protein binding. When fusion peptides or loops insert into the membrane, they create voids beneath the protein. Polyunsaturated lipids may fill these voids by inserting their flexible acyl chains, while their headgroups fit into binding pockets. This dual role of binding and void-filling is not possible with saturated or shorter-chain lipids. In addition to these roles, Pinot et al.[56], showed that polyunsaturated lipids (C18:0–C22:6) induce shallow defects in the membrane that fusion peptides or loops can exploit, further enhancing their binding affinity. Notably, we recently found from CG simulations that polyunsaturated lipids favor fusion by greatly reducing the free energy cost of forming a stalk-like fusion intermediate[4]. Thus, we hypothesize that protein-bound polyunsaturated lipids play a pivotal role not only in the initial binding of fusion proteins to the host membrane but may also stabilize the stalks along the fusion pathway.

The binding affinities of Gc and gB are weaker by tens of kilojoule per mole compared to class I fusion proteins, suggesting that class II and III fusion proteins may utilize different mechanisms for binding to host membranes. Gc has two distinct lipid binding pockets: one on individual monomers and another at the monomer–monomer interface. The monomer-specific lipid binding pocket is accessible in the intermediate state, where individual monomers bind to the host membrane before assembling into trimers[12,13,47]. In contrast, gB lacks a monomer-specific lipid binding pocket. Instead, the fusion loops, which are widely spaced at the viral membrane, form in the fusogenic trimeric state a lipid binding pocket through a unique fusion loop crossover at the monomer–monomer interface[16]. It will be highly interesting to address whether the lipid binding pockets may be targeted by ligands with the aim to interfere with membrane binding by fusion proteins. We speculate that, in addition to binding via the lipid binding pocket, the membrane binding may also be stabilized by glycoproteins involved in the gB fusion pathway, although their exact role remains unclear. These differences in lipid binding pockets likely reflect the distinct maturation pathways of class II and III fusion proteins.

We furthermore highlight the role of gangliosides as host attachment factors during viral infection. Our simulations show an enhanced binding affinity of fusion proteins in the presence of GM3, which we attribute to the non-specific, broad distributions of GM3 around the fusion proteins. This suggests that ganglioside sugar headgroups may serve as initial attachment points for fusion proteins before they engage with the host membrane. These findings align with experimental data showing the co-localization of

pseudorabies virus (PrV) with gangliosides[32], and in this study, we demonstrate the association of its fusion protein, gB, with these lipids. This initial binding of gangliosides is reminiscent of their role in membrane targeting by bacterial toxins, such as shiga toxin[57] and cholera toxin[58], which also rely on gangliosides for initial attachment before subsequent interactions with the host membrane.

In summary, our study provides energetic and structural insights into the membrane-binding affinities of viral fusion proteins from classes I, II, and III. IAV HA2 exhibits strong membrane binding with free energy values < −100 kJ/mol, suggesting essentially irreversible binding, in contrast to by far weaker affinities of Gc and gB. Membrane binding is co-regulated by cholesterol and polyunsaturated lipids, with Gc and gB utilizing unique lipid-binding pockets to engage with the host membrane. The significance of lipid-binding pocket of Gc was validated by a point mutation that reduced its membrane-binding affinity, while the affinity was rescued by the presence of anionic PS lipids. These differences in membrane binding among fusion proteins reflect their distinct maturation pathways. Additionally, we demonstrate that gangliosides may serve as initial attachment points for fusion proteins prior to full membrane engagement. These findings deepen our understanding of lipid functions in viral infection, signaling, and transport, as previously reported[52,59–64]. Given that many viruses alter lipid synthesis upon entry and during assembly[40,41], our work underscores the pivotal role of virus–lipid interactions in viral replication.

## Methods

### Coarse-grained simulations and analysis

The lipid environment of viral fusion proteins and the binding free energies of viral protein-membrane interactions were studied using Martini coarse-grained (CG) MD simulations[65,66]. To model interactions of fusion proteins with the PM outer leaflet, we used symmetric membrane models, however, with the composition taken from the PM outer leaflet. This setup provided a realistic composition of the protein-interacting leaflet, while avoiding putative complications owing differential bilayer stress, which may be present in asymmetric membrane models. For cholesterol, we used the reparametrized Martini model, which reproduced lipid phase behavior in mixed membranes[67]. For gangliosides, we used a modified Martini parameterization designed to capture ganglioside clustering observed at the atomistic level[68]. In addition, to better estimate the contribution of individual lipid species to binding energies, we furthermore performed simulations with membranes composed of two or three lipid types. Tables S1–S3 list the simulated membrane compositions with varying headgroup type, tail length, degree of acyl tail unsaturation. As for the structure of post-fusion viral proteins, class I was represented using influenza A virus hemagglutinin (HA2) (Protein Data Bank ID: 1QU1)[19] with the fusion peptide modelled by an NMR structure (PDB ID: 1IBN)[69]. For class II, we used the crystal structure of Rift Valley fever virus Gc (Protein Data Bank ID: 6EGU)[22] in wild-type form. For mutation studies, the D961K mutant was modeled using CHIMERA[70]. For class III, the crystal structure of pseudorabies virus gB (PDB ID: 6ESC)[25] was used. Missing loops were modeled using MODELLER[71]. The MARTINI 3(beta) force field was used to describe the CG systems[66]. To set up the CG simulation systems, the atomistic protein structures were converted to CG models using the martinize script[65], while the secondary and tertiary structure were preserved using elastic networks with a elastic bond force constant of 500kJ/mol · nm² and elastic bond cut-offs set to 0.5 nm (lower) and 0.9 nm (upper). The linker to the fusion peptide of HA2 was flexible. While we cannot exclude that the elastic network suppressed conformational fluctuations of the protein, the agreement with a wide body of experimental data suggests that the elastic network restraints do not affect the key conclusions of this study (see Discussion). The protein–membrane systems were prepared using the insane.py script with the protein positioned slightly above the membrane surface[72]. The system was solvated with non-polarizable MARTINI CG water and neutralized with NaCl beads. The initial configuration was energy minimized

using the steepest-descent algorithm to remove bead clashes. During equilibration, a cut-off of 1.1 nm was used for Lennard-Jones and Coulombic interactions. Bond lengths were constrained using LINCS[73]. The temperature of 310 K was controlled using velocity rescaling with a coupling constant of 1.0 ps and the pressure of 1 bar was controlled with the semi-isotropic Parrinello-Rahman[74,75] barostat with a coupling contant of 12 ps. All simulations were carried out with a time step of 20 fs using GROMACS version 2020.2[76].

The protein–membrane systems listed in Tables S1 and S2 were simulated in triplicates, with a total simulation time of 45 μs per membrane composition. For analysis, last 10 μs of each repeat was concatenated to yield a trajectory of 30 μs per membrane composition. The long equilibration times of 15 μs ensured that lipid headgroups were bound to lipid binding sites. The lipid enrichment maps shown in Fig. 2B and Fig. S2–S5 were obtained by first computing the lipid density using gromacs tool gmx densmap, and then converted to enrichment level[59]. Similarly, the lipid enrichment index shown in Fig. 2C is described below using the approach outlined by Corradi et al.[59]. We compare the lipid concentration $c_{prot}$ within 0.65 nm from the protein with the lipid concentration $c_{bulk}$ in the bulk membrane. The enrichment index is defined as $c_{prot}/c_{bulk}$. Here, the gromacs tool gmx select was used to obtain the lipid concentration within 0.65 nm from the protein. For density calculations, lipid phosphate group (PO4 bead) for PC, PE, PS and SM lipids; the hydroxyl group (ROH bead) for cholesterol and AM1 bead for ganglioside GM3 were used. For the computation of contact maps, a contact was recorded if the distance between cholesterol and protein residues was within 0.65 nm. Multiple sequence alignment and sequence identity matrices were generated using Clustal Omega[77]. All images and plots were generated using VMD[78] and matplotlib[79].

### Calculation of binding free energies

The free energy of binding was obtained from potentials of mean force (PMF) computed with umbrella sampling (US) simulations. The last frame (protein bound to membrane state) from conventional equilibration simulation served as starting structure for US simulations. The reaction coordinate was defined as the distance between the center of mass (COM) distance along z-direction (membrane normal) between the protein and the phosphate (PO4) beads of the upper, protein-bound membrane leaflet. Here only PO4 beads within the cylinder were used to compute the membrane COM, where the cylinder was aligned along the z-axis with the axis at the protein COM. The pull-cylinder-r parameter was set to 2.5 nm. This modification of the reaction coordinate avoids that the PMF is smeared out due to membrane bending, while all protein-interacting lipids are taken into account for defining the membrane center of mass. Steered MD simulations to generate frames for US simulations were performed in two stages. First, the protein was further pulled into the membrane by 0.5 nm over 50 ns with a force constant of $k = 4000$ kJ mol⁻¹ nm⁻² and a pull rate of $1 \cdot 10^{-5}$ nm ps⁻¹. Next, the protein was pulled away from the membrane surface by 3.5 nm to 4.5 nm over 240 ns, depending on the protein class and lipid composition, with a force constant of $k = 1000$ kJ mol⁻¹ nm⁻² and a pull rate of $1.5 \cdot 10^{-5}$ nm ps⁻¹. Starting frames for US were taken from the steered MD simulations, with neighboring US windows spaced at 0.05 nm when the protein was interacting with the membrane and 0.1 nm when protein was in the bulk water. Each window was simulated for 500 ns, while the last 400 ns were used to calculate the PMFs using the weighted histogram analysis method[80], as implemented by the gmx wham code[81]. Statistical errors were estimated using 50 rounds of Bayesian bootstrapping of complete histograms[81]. Furthermore, convergence was assessed by binning all umbrella histograms into 100 ns time blocks. In agreement with the bootstrapping analysis, the uncertainty of the PMFs is within a few kilojoules per mole (Fig. S12). The free energy of the unbound state (at large protein–membrane distance) was defined to zero. Hence, the binding free energies $\Delta G_{bind}$ were taken from the free energy minimum of the PMFs.

## Atomistic simulations

To obtain atomistic insight into lipid interactions with RVFV Gc and PrV gB, we backmapped equilibrated coarse-grained systems to atomistic resolution and carried out all-atom MD simulations with a total simulation time of 40 $\mu$s. For resolution transformation, we selected systems with varying headgroups (PC, PS, PE and GM3) and lipid tail unsaturation. Table S4 lists the simulated systems. The resolution transformation from CG to all-atom was carried out using the Backward tool[82]. Next, protein crystal structures were superposed onto the structure obtained from Backward. The proteins were glycosylated using the core mannose sugars (GlcNAc$_2$Man$_3$) using doGlycans[83] and later processed with the CHARMM-GUI Glycan Reader & Modeler tool[84]. Table S4A lists the CG systems backmapped to all-atom along with the membrane composition and glycosylation sites. The glycosylated protein-membrane systems were solvated with TIP3P water[85] and neutralized with NaCl ions. Additional 150 mM NaCl were added to mimic the physiological salt concentration. The systems was energy minimized using the steepest-descent algorithm, followed by several rounds of short equilibration with protein backbone restrained. The CHARMM36m force field was applied for protein and CHARMM36 for lipids[86–88], using corrections for cation-$\pi$ interactions by Khan et al.[89]. Lennard-Jones forces were gradually switched off between distances of 1 nm and 1.2 nm. Coulombic interactions were treated with the particle-mesh Ewald method[90,91] with a real-space cutoff at 1.2 nm. Bonds involving hydrogen atoms were constrained with LINCS[73]. The geometry of water molecules was constrained with SETTLE[92]. The temperature of 310 K was controlled using velocity rescaling[93] with a coupling constant of 1.0 ps. The pressure of 1 bar was controlled with the semiisotropic Parrinello-Rahman barostat[74,75] with a coupling constant of 5 ps. The final production run was performed without restraints. Each simulation was run for 2 $\mu$s with a time step of 2 fs using GROMACS version 2020.2[76]. Figures 5 and 6 show snapshots after 2 $\mu$s of simulation for PAPC or PAPS binding to RVFV Gc as well as for PAPC or GM3 binding to PrV gB, respectively. Similar interactions were found for POPC or PAPE (Figs. S8–S9). All figures were rendered using VMD[78].

**Statistics and reproducibility.** For the data presented in (Fig. 2B,C), simulations were performed in triplicates with different initial velocities. For the data presented in (Figs. 2D, E, 4B, S1 and S10) convergence of PMFs was shown using Bayesian bootstrapping of complete histograms, which provides a conservative error estimate. Convergence was furthermore confirmed by binning all umbrella simulations into 100 ns time blocks[81].

## Reporting summary

Further information on research design is available in the Nature Portfolio Reporting Summary linked to this article.

## Data availability

All data are available in the main text and/or in the supplementary information pdf file. Source data for the graphs are provided as a Supplementary Data file with this paper. Coarse-grained and all-atom structure files, topologies, parameter files, backmapping files, and other related files required to reproduce the simulation data are available at Zenodo: https://doi.org/10.5281/zenodo.7945458.

## Code availability

All software and analysis tools used in this study are publicly available and detailed in the Methods section. No custom code was developed.

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

## Acknowledgements

We thank Marija Backovic for critical reading of this manuscript. **Funding:** This study was supported by the Deutsche Forschungsgemeinschaft via SFB 1027/B7.

## Author contributions

Conceptualization: C.S.P., J.S.H. Methodology: C.S.P., J.S.H. Investigation: C.S.P., T.B. Visualization: C.S.P., T.B. Supervision: J.S.H. Writing-original draft: C.S.P. Writing-review & editing: C.S.P., J.S.H.

## Funding

## Competing interests

The authors declare no competing interests.

## Additional information

**Peer review information** : *Communications Biology* thanks Zeinab Rahimi and the other, anonymous, reviewers for their contribution to the peer review of this work. Primary Handling Editors: Yun Luo and Laura Rodriguez Perez. A peer review file is available.

