## [Transparent Peer Review file · Communications Biology]

Viral fusion proteins of classes II and III recognize and reorganize complex biological membranes

Corresponding Author: Professor Jochen Hub

Version 0:

Reviewer comments:

Reviewer #1

(Remarks to the Author)

The manuscript describes simulations of 3 viral fusion proteins, from each of the 3 main families of viral fusion proteins, with the goal of clarifying the role of protein-lipid interactions in determining binding to the host membrane. The subject is of high and general interest in biology. The manuscript is very clear in most parts.

The methodology is very solid and aligns with the state-of-the-art in molecular simulations.

I only have a few minor suggestion that could hopefully contribute to enhance clarity.

- It appears that the structures used in all the CG simulations are representative of the post-fusion state, is that correct? I am not a specialist in the field, so I wonder why it is relevant to study binding of the post-fusion conformation, instead of pre-fusion. This should be clarified.
- The statements on the differences in binding affinities among the 3 different classes of viral fusion proteins are based on sequence conservation of relevant binding regions. Not being an expert in this field, I wonder if an assessment of overall sequence conservation is sufficient for such broad statement, particularly for class I proteins, where conservation is weak; in general, a single point mutation is sufficient to alter the binding affinity by tens of kJ/mol. I would recommend clarifying this point further.
- The assessment of the protein-lipid interactions relevant for membrane binding seems rather qualitative: a number of specific hydrogen bonds and cation- π interactions are listed, but it is difficult to grasp how stable (and hence, how important) those interactions are. I would recommend showing some quantitative assessment of those specific interactions.
- The description of protein-membrane interaction for the class III protein makes reference to a comparison between the structure of the fusion loop in the pre- and post-fusion states. However, this remains difficult to understand as such structures are not shown. Adding the structures would clarify the point.
- It seems that protein structures were modelled at the CG level in the absence of glycans, while at the all-atom level they were glycosylated. In both cases, glycans do not interact with the membrane, is that correct? Why were they included in the modeling? Do they affect the conformation of the protein?

Reviewer #2

(Remarks to the Author)

This manuscript presents a comprehensive computational study of the membrane-binding interactions of class I, II and III viral fusion proteins (HA2, Gc and gB) using coarse-grained molecular dynamics (CGMD) simulations followed by umbrella sampling (US) to calculate binding free energies. This study could be useful in designing antiviral therapies by understanding how these proteins interact with the membrane. While the study provides valuable insight into the role of lipids in viral fusion, several aspects of the computational methodology need to be addressed in more detail.

I suggest the following minor changes before publication:

1. The authors mention the use of elastic networks to preserve protein structure in CG models, but do not specify: The force constants for the elastic network restraints. How were these restraints optimized for each fusion protein class, and how do they affect the conformational flexibility of the proteins, particularly during lipid binding?
2. The equilibration steps are described, but details are lacking, such as: how long were the protein and membrane equilibration phases, and how was stability confirmed during these steps?
3. How the elastic network restraints affect protein structure and flexibility at a coarse-grained level. How do these restraints influence the protein's flexibility and its interaction with the membrane, especially when compared to all-atom simulations?
4. The reaction coordinate for PMFs includes a cylindrical pull with a 2.5 nm radius. Was this parameter optimized for these systems, and how sensitive are the results to this choice? Would a broader or narrower pull region influence the calculated binding affinities?
5. Given the importance of PMF convergence for accurate free energy calculations, how was the convergence of the umbrella sampling windows verified, particularly for membranes with different lipid compositions?
6. While the study provides computational details, the biological and therapeutic implications are underexplored. For example, how might the identified lipid-binding mechanisms be targeted pharmacologically? Expanding the discussion of the physiological relevance of lipid specificity (e.g. PS in endosomal membranes, cholesterol in plasma membranes) would strengthen the manuscript.

Reviewer #3

(Remarks to the Author)

In this study, the authors performed a systematic evaluation of how different classes of fusion peptides bind to lipid membranes, based on high throughput molecular dynamics simulations. In particular the role of individual lipid types in modulating the binding is being investigated. Overall the study is well designed and performed, and the results are interesting, providing novel insights on the interplay between the fusion machinery and lipid membrane components.

I have a few comments that could further strengthen the paper:

- 1) Only one representative from each fusion class was used. Although I understand that the computational work is already quite impressive, simulation of other representatives from the different classes would provide more confidence in the generic conclusions. These could focus on selective systems (e.g. probing the role of polyunsaturated lipids or cholesterol, or comparing binding PMFs for a given membrane composition). To simplify the calculations, one could even focus on the fusion peptides instead of the entire protein.
- 2) PMFs of protein-lipid interactions are notoriously difficult to converge. Although error estimates are presented, I would urge the authors to also present evidence of convergence of the profiles such as shown in Figure 2E. Dividing the trajectory into smaller blocks, or comparing between the replicates would be a suitable way of doing so.
- 3) The PM model was built symmetrically if I am not mistaken, but real PM membranes are asymmetric. This choice should also be explained and discussed in more detail, as it could have implications for the findings in this work.
- 4) The authors conducted a nice sequence analysis on the different classes of fusion proteins, but given the importance of these proteins I wonder whether such an endeavour has been done before by other research groups ?
- 5) Exceptions to the order of binding affinity are mentioned, referring to Tables S5-S7, but I cannot infer that from this data. Please be more precise.
- 6) Uncertainty estimates are lacking for the reported binding free energies throughout the manuscript.
- 7) The PMFs shown in Figure 4A seem to have an entry barrier. This warrants some explanation.
- 8) In the Discussion section, the authors argue that the strong membrane binding energy (115 kJ/mol) of a single trimer implies that three trimers are sufficient to overcome the energy required for stalk formation (ranging between 250-500 kJ/mol). I do not see how you can meaningfully compare these two different energies. For instance, a membrane spanning protein surely has an even much stronger binding energy to the membrane, but will likely fail to favour the formation of stalks.
- 9) Some essential details on how the modelling was performed are missing. For instance, how were disordered protein loops in Martini treated (as Martini protein models depend on assignment of secondary structural elements)? What was the force field used for cholesterol, and for the gangliosides (different models have been put forward by the Martini community, the choices should be discussed in light of the advantages/disadvantages of these models) ?
- 10) Moreover, apparently a beta version of the Martini 3 model was used - is there a link to find the actual parameters? A

proper citation of the martinize tool is also lacking, this tool has been published in eLife already a while ago. Finally, elastic networks are mentioned but it is unclear what this entails.

Version 1:

Reviewer comments:

Reviewer #1

(Remarks to the Author)

The authors addressed the concerns by the reviewers to a sufficient extent.

Reviewer #2

(Remarks to the Author)

Dear professor Luo,

I have reviewed the revised manuscript entitled "Viral fusion proteins of classes II and III recognize and reorganize complex biological membranes" by Professor Hub and colleagues, and I am happy to confirm that it is now suitable for publication. The revisions have addressed the concerns raised in the previous review, and I believe the manuscript is now ready to proceed.

Best regards,
Zeinab Rahimi

Reviewer #3

(Remarks to the Author)

The manuscript has been nicely revised, all my comments have been adequately addressed.

The only thing missing is proper citation of the cholesterol and ganglioside models used.

Reviewers' comments:

Reviewer #1 (Remarks to the Author):

The manuscript describes simulations of 3 viral fusion proteins, from each of the 3 main families of viral fusion proteins, with the goal of clarifying the role of protein-lipid interactions in determining binding to the host membrane. The subject is of high and general interest in biology. The manuscript is very clear in most parts. The methodology is very solid and aligns with the state-of-the-art in molecular simulations. I only have a few minor suggestions that could hopefully contribute to enhance clarity.

REPLY: Thank you for the positive feedback and for the valuable suggestions and critiques. Please find below or point-by-point response.

- It appears that the structures used in all the CG simulations are representative of the post-fusion state, is that correct? I am not a specialist in the field, so I wonder why it is relevant to study binding of the post-fusion conformation, instead of pre-fusion. This should be clarified.

REPLY: In the pre-fusion structures, the fusion proteins are still embedded at the viral membrane and are not yet extended toward the host membrane. The reviewer probably wonders why we did not simulate an intermediate state, in which the fusion protein points towards the host membrane, while still being anchored in the viral membrane. The reason is that the intermediate is unstable and transient, so it is difficult to obtain structural data, whereas the postfusion state is an energetic minimum and therefore accessible to structural characterization.

However, the protein-membrane interface of the -structurally characterized- postfusion structures are similar to the interface in the intermediate, except that the transmembrane anchors are in the postfusion structure located next to the protein-host membrane interface (and not in the viral membrane anymore). Thus, we use the postfusion structure with the transmembrane anchors removed (which are not resolved anyway by crystallography) as a model for the intermediate.

To make this point clearer, we added to the Results (lines 155 - 160):

“To model the interactions of the viral fusion proteins with host membranes prior to the collapse of the fusion proteins, we simulated the structurally well-characterized post-fusion conformations in the absence of their transmembrane helical anchors.”

- The statements on the differences in binding affinities among the 3 different classes of viral fusion proteins are based on sequence conservation of relevant binding regions. Not being an expert in this field, I wonder if an assessment of overall sequence conservation is sufficient for such broad statement, particularly for class I proteins, where conservation is weak; in general, a single point mutation is sufficient to alter the binding affinity by tens of kJ/mol. I would recommend clarifying this point further.

Thank you for raising this point. Our proposed generalization of binding affinities is based on the presence or conservation of hydrophobic residues that bind to the membrane, rather than on the overall sequence similarity. Class I proteins reveal a larger fusion peptide with multiple hydrophobic groups (Table S5), suggesting that other Class I proteins reveal similarly large binding affinities like the HA2 protein studied here. In contrast, Class II and III proteins exhibit by far smaller fusion loop, which contain mostly 2 or only 1 bulky hydrophobic residue (see Tables S6 and S7), although exceptions exist. Thus, these trends suggest that the trends in binding affinities follow roughly Class I > Class II ≥ Class III.

Having said this, we fully agree that the text was not sufficiently clear. Thus, we now write more carefully (lines 227 - 235):

"Thus, although we computed binding affinities in this study only for one representative of each class, the sequence analysis suggests that the trends in binding affinities obtained for HA₂, Gc, and gB likely apply for many other members of the fusion protein classes. Specifically, the strong binding observed for IAV HA₂ is expected..."

And we now write more carefully (lines 194 - 198): "However, similar to IAV HA₂, fusion peptides of all class I fusion proteins contain several hydrophobic amino acids, suggesting that many class I fusion proteins may bind to the host membrane with high affinity."

- The assessment of the protein-lipid interactions relevant for membrane binding seems rather qualitative: a number of specific hydrogen bonds and cation- π interactions are listed, but it is difficult to grasp how stable (and hence, how important) those interactions are. I would recommend showing some quantitative assessment of those specific interactions.

We agree that assigning variations in membrane binding affinity to individual residues would be insightful. However, this would require a large set of additional PMF calculation with proteins with single-point mutations (e.g. a computational alanine scan), followed by quite detailed discussions. Since the study is already quite rich in data, we feel that the study would lose focus with such discussion. Instead, to best serve the reader, we focus on a mutation for which experimental binding data is available (D961K mutant of RVFV Gc, Fig. S10) and to a structural characterization of the lipid-protein interactions (Figs. 5 and 6)

- The description of protein-membrane interaction for the class III protein makes reference to a comparison between the structure of the fusion loop in the pre- and post-fusion states. However, this remains difficult to understand as such structures are not shown. Adding the structures would clarify the point.

Thank you for this suggestion. We have now included representative structures of pre- and postfusion states for class III fusion proteins in a new Fig. S11 and referenced in the manuscript (line 501).

- It seems that protein structures were modelled at the CG level in the absence of glycans, while at the all-atom level they were glycosylated. In both cases, glycans do not interact with the membrane, is that correct? Why were they included in the modeling? Do they affect the conformation of the protein?

Glycans linked to fusion proteins do not interact with the membrane, and their role in membrane binding and fusion remains unclear. Our simulations did not reveal any significant effect of glycans on protein conformation. However, in our atomistic simulations, we included glycans to model the protein as closely as possible to experimental conditions, as they are an integral part of fusion proteins. Additionally, these glycosylated structures have been uploaded to Zenodo (see *Data and Materials Availability*) to support future simulation studies, should a connection between glycans and the fusion pathway be established.

Reviewer #2 (Remarks to the Author):

This manuscript presents a comprehensive computational study of the membrane-binding interactions of class I, II and III viral fusion proteins (HA₂, Gc and gB) using coarse-grained molecular dynamics (CGMD) simulations followed by umbrella sampling (US) to calculate binding free energies. This study could be useful in designing antiviral therapies by understanding how these proteins interact with the membrane. While the study provides valuable insight into the role of lipids in viral fusion, several aspects of the computational methodology need to be addressed in more detail.

REPLY: Thank you for reviewing our manuscript and detailed critiques. Please find our point-by-point response below.

I suggest the following minor changes before publication:

1. The authors mention the use of elastic networks to preserve protein structure in CG models, but do not specify: The force constants for the elastic network restraints. How were these restraints optimized for each fusion protein class, and how do they affect the conformational flexibility of the proteins, particularly during lipid binding?

Thank you for spotting this, as details on the elastic network has indeed been missing. We applied an elastic network with a bond force constant of 500 kJ/mol-nm² and set the lower and upper elastic bond cut-offs to 0.5 nm and 0.9 nm, respectively. These parameters, now included in the Methods section (lines 730–734):

“elastic networks with a elastic bond force constant of 500 kJ/mol-nm², and elastic bond cut-offs set to 0.5 nm (lower) and 0.9 nm (upper). The linker to the fusion peptide of HA2 was kept flexible”.

Based on our extensive validation against experimental data, it is unlikely that the elastic network restraints affect the conclusions of this study, but that instead the MARTINI model together with the elastic network provides a reasonable approximation. This is implied by (i) the correct identification of the crystallographic lipid binding pocket (Ref. 22, Guardado-Calvo *et al.*, *Science* 2017), (ii) correct trends in ΔG_{bind} values with increasing cholesterol content in agreement with Refs. 22 and 25 (Vallbracht *et al.*, *J Virol* 2018), (iii) correct trends in binding affinities of D961K mutant of RVFV Gc in response to phosphatidylserine (PS) lipids (Ref. 22). (iv) ΔG_{bind} values of the IAV HA2 fusion peptide in excellent agreement with experimental data (Fig. S1).

Having said this, we cannot exclude that the elastic network restraints have a *quantitative* effect on binding affinities. Without elastic network, the protein would yield increased flexibility or even partly unfold, which would likely yield stronger (more negative) ΔG_{bind} values. However, whether such increased flexibility is realistic or a consequence of the approximations underlying the MARTINI model, would be unclear. Therefore, our strategy used here is (i) to restrain the coarse-grained structure to the crystallographic conformation, (ii) use extensive validation against experimental data, and (iii) use backmapping to all-atom resolution to validate that key interactions are maintained with more quantitative all-atom simulations.

We now added a paragraph to the Discussion (lines 561 - 574): “We computed the free energy of membrane binding for three viral fusion proteins using coarse-grained simulations, and we validated the simulations against a wide body of experimental data: the simulations...”

And we now write in the Methods (lines 734 – 739): “While we cannot exclude that the elastic network suppressed conformational fluctuations of the protein, the agreement with a wide body of experimental data suggests that the elastic network restraints do not affect the key conclusions of this study (see Discussion).”

2. The equilibration steps are described, but details are lacking, such as: how long were the protein and membrane equilibration phases, and how was stability confirmed during these steps?

The protein-membrane systems were equilibrated for 15 microseconds, as stated in line 760 of the manuscript. Stability was assessed by visualizing the binding of the protein to the membrane, which remained stable throughout the equilibration phase. This was further confirmed by lipid enrichment maps shown in Fig 2B,C and S2-S5, demonstrating that membrane binding facilitates lipid recruitment to the binding pocket. To rationalize these long equilibration times, we added (lines 760 - 761):

“The long equilibration times of 15 μ s ensured that lipid headgroups were bound to lipid binding sites.”

3. How the elastic network restraints affect protein structure and flexibility at a coarse-grained level. How do these restraints influence the protein's flexibility and its interaction with the membrane, especially when compared to all-atom simulations?

Regarding the validity of elastic network restraints for ΔG_{bind} values predictions, please see our response to question (1.) above.

Regarding the flexibility of the protein, the fusion peptide of HA2 (class I) was connected with the rest of the protein with a flexible linker during CG simulation, rationalizing the dispersed lipid densities shown in Fig. S2.

See lines added to discussion (lines 561 - 574) and Methods (lines 734 - 739).

4. The reaction coordinate for PMFs includes a cylindrical pull with a 2.5 nm radius. Was this parameter optimized for these systems, and how sensitive are the results to this choice? Would a broader or narrower pull region influence the calculated binding affinities?

The radius defines the lateral region of the membrane that is used to compute the z position of the membrane center of mass. This choice ensures that large-scale membrane undulations would not lead to a smearing out of the reaction coordinate (here taken as the center-of-mass distance between protein and the cylindrical membrane region below the protein). Here, the choice of a 2.5 nm radius was based on the approximate protein–membrane interaction region, thereby taking all protein-interacting lipids (including lipids bound to lipid binding sites) into account for computing. A smaller radius could exclude essential lipid interactions, while a (far) larger radius could trigger undesired membrane undulations. We now rationalize this choice more precisely (lines 797 - 801):

“This modification of the reaction coordinate avoids that the PMF is smeared out due to membrane bending, while all protein-interacting lipids are taken into account for defining the membrane center of mass.”

5. Given the importance of PMF convergence for accurate free energy calculations, how was the convergence of the umbrella sampling windows verified, particularly for membranes with different lipid compositions?

Thank you for raising this point. We assessed the convergence of the PMFs by computing the statistical uncertainty by bootstrapping complete histograms, which typically gives a more conservative uncertainty estimate compared to techniques such as time binning analysis, since bootstrapping complete histograms does not require knowledge of the (typically unknown) autocorrelation times. These uncertainties are shown in all PMFs as shaded areas (Fig. 2E, 4A, S1A, S10A) and as vertical bars in ΔG_{bind} values (Fig. 2D, 4B, S1B, S10B)

However, in response to the reviewer, now carried out in addition a binning analysis for Gc bound to the complex plasma membrane by binning all umbrella histograms into 100ns blocks. In agreement with the bootstrapping analysis, the uncertainty is in the range of few kilojoules per mole (see new Fig. S12 and lines 821 - 825):

“Furthermore, convergence was assessed by binning all umbrella histograms into 100ns time blocks. In agreement with the bootstrapping analysis, the uncertainty of the PMFs is within few kilojoules per mole.”

6. While the study provides computational details, the biological and therapeutic implications are underexplored. For example, how might the identified lipid-binding mechanisms be targeted pharmacologically? Expanding the discussion of the physiological relevance of lipid specificity (e.g. PS in endosomal membranes, cholesterol in plasma membranes) would strengthen the manuscript.

We fully agree, targeting the lipid binding site with a ligand would be an exciting direction of future research, but this is beyond the scope of this study. We added:

“It will be highly interesting to address whether the lipid binding pockets may be targeted ligands with the aim to interfere with membrane binding by fusion proteins.”

Regarding the distribution of lipids, we write (lines 427 - 430):

“The physiological relevance of binding to PS lipids aligns with Gc entry via the endocytic pathway, where endosomal membranes are rich in PS and other anionic lipids.”

And (lines 255 - 260):

“Viral fusion proteins target the outer leaflet of the host PM, which is enriched with PC lipids of varying tail compositions and cholesterol [39]. PC headgroups and cholesterol play a functional role during viral infection as shown by previous experiments.”

Reviewer #3 (Remarks to the Author):

In this study, the authors performed a systematic evaluation of how different classes of fusion peptides bind to lipid membranes, based on high throughput molecular dynamics simulations. In particular the role of individual lipid types in modulating the binding is being investigated. Overall the study is well designed and performed, and the results are interesting, providing novel insights on the interplay between the fusion machinery and lipid membrane components.

REPLY: Thank you for the positive feedback and for the careful evaluation of our manuscript.

I have a few comments that could further strengthen the paper:

1) Only one representative from each fusion class was used. Although I understand that the computational work is already quite impressive, simulation of other representatives from the different classes would provide more confidence in the generic conclusions. These could focus on selective systems (e.g. probing the role of polyunsaturated lipids or cholesterol, or comparing binding PMFs for a given membrane composition). To simplify the calculations, one could even focus on the fusion peptides instead of the entire protein.

Many thanks for these suggestions. Regarding the suggestion of simulating additional fusion peptides only, we would face the problem that binding affinities of three peptides do not add up to the binding affinity of a fusogenic trimer. This discrepancy likely arises because the binding of the entire fusion protein induces a larger perturbation of the membrane compared to the binding of three individual fusion peptides. Thus, the implications may be limited and would require extensive discussions, which we would like to avoid for the sake of focus.

Regarding the generic conclusions based on PMF calculations of one representative each class, we agree that PMFs with additional fusion proteins would be insightful. In this study, however, we followed a different strategy with the aim to provide a more holistic view on the membrane-interacting residues of viral fusion proteins. We based our conclusions on the presence or conservation of hydrophobic residues that bind to the membrane. Class I proteins reveal a larger fusion peptide with multiple hydrophobic groups (Table S5), suggesting that other Class I proteins reveal similarly large binding affinities like the HA₂ protein studied here. Class II and III proteins exhibit fusion loops with mostly 2 or only 1 bulky hydrophobic residue (see Tables S6 and S7), although exceptions exist. Thus, these trends suggest that the trends in binding affinities follow roughly Class I > Class II \approx Class III.

However, we agree that a more careful statement is appropriate here. We now write (lines 227 - 235):

“Thus, although we computed binding affinities in this study only for one representative of each class, the sequence analysis suggests that the trends in binding affinities obtained for HA₂, Gc, and gB likely apply for many other members of the fusion protein classes. Specifically, the strong binding observed for IAV HA₂ is expected...”

And we now write more carefully (lines 194 - 198): “However, similar to IAV HA₂, fusion peptides of all class I fusion proteins contain several hydrophobic amino acids, suggesting that many class I fusion proteins may bind to the host membrane with high affinity.”

2) PMFs of protein-lipid interactions are notoriously difficult to converge. Although error estimates are presented, I would urge the authors to also present evidence of convergence of the profiles such as shown in Figure 2E. Dividing the trajectory into smaller blocks, or comparing between the replicates would be a suitable way of doing so.

We assessed the convergence of the PMFs by computing the statistical uncertainty by bootstrapping complete histograms, which typically gives a more conservative uncertainty estimate compared to techniques such as time binning analysis, since bootstrapping complete histograms does not require knowledge of the (typically unknown) autocorrelation times. These uncertainties are shown in all PMFs as shaded areas (Fig. 2E, 4A, S1A, S10A) and as vertical bars in ΔG_{bind} values (Fig. 2D, 4B, S1B, S10B)

However, in response to the reviewer, now carried out in addition a binning analysis for Gc bound to the complex plasma membrane by binning all umbrella histograms into 100ns blocks. In agreement with the bootstrapping analysis, the uncertainty is in the range of few kilojoules per mole (see new Fig. S12 and lines 821 - 825):

“Furthermore, convergence was assessed by binning all umbrella histograms into 100ns time blocks. In agreement with the bootstrapping analysis, the uncertainty of the PMFs is within few kilojoules per mole.”

3) The PM model was built symmetrically if I am not mistaken, but real PM membranes are asymmetric. This choice should also be explained and discussed in more detail, as it could have implications for the findings in this work.

REPLY: Yes, we simulated a symmetric model of the plasma membrane, however, with the lipid composition of the outer leaflet. We now write more explicitly (lines 701 - 708):

“To model interactions of fusion proteins with the PM outer leaflet, we used symmetric membrane models, however, with the composition taken from the PM outer leaflet. This setup provided a realistic composition of the protein-interacting leaflet, while avoiding putative complications owing differential bilayer stress, which may be present in asymmetric membrane models.”

Since the fusion proteins mostly interact with the upper leaflet in our simulations, this setup provides a reasonable model for computing binding affinities.

4) The authors conducted a nice sequence analysis on the different classes of fusion proteins, but given the importance of these proteins I wonder whether such an endeavour has been done before by other research groups?

REPLY: In fact, we were also surprised that such analysis has, to best of our knowledge not been carried out before. Some multiple sequence alignments have been shown with a focus on conservation of fusion loops (Ref. 22, 25), and sequence analysis is (of course) extensively used to study evolution of specific virus species, e.g. during pandemics. However, a systematic sequence analysis among viral fusion proteins has, to the best of our knowledge, not been available before.

5) Exceptions to the order of binding affinity are mentioned, referring to Tables S5-S7, but I cannot infer that from this data. Please be more precise.

Thank for pointing this out, as the manuscript has indeed been unclear at this point. We have changed the section in the Results and emphasize more clearly, that the trend in binding affinities class I > class II \geq class III is merely a trend but not a strict rule, as clear exceptions exist. Therefore, we changed the section title to “Membrane binding affinities follow the order class I > class II \geq class III for many fusion proteins” and write in the paragraph (lines 204 - 226):

“In agreement with RVFV Gc, many of the class II fusion loops analyzed here contain one tryptophan and at one phenylalanine that may insert into the membrane, although several variations exist : fusion loops of togaviridae contain a second phenylalanine, while in fusion loops of certain phenuiviridae tryptophan is replaced with a hydrophobic leucine or isoleucine (Table S2, see underlined Phe/Trp residues). Among class III, fusion proteins exhibit a low degree of conservation with an exception for the baculoviridae family. Notably, albeit low

conservation between the complete fusion proteins of baculoviridae and some genera within orthomyxoviridae families, the matrix reveals high conservation between the fusion loops of these families, indicating their evolutionary relation (Fig. 3C, central blocks). Among the class III fusion proteins analyzed here, baculoviridae, orthomyxoviridae, and several herpesviridae contain only one hydrophobic aromatic residue (Phe/Trp) per monomer to insert into the membrane (Table S3). However, variations to this observation exist, as shown by Epstein-Barr virus or several rhabdoviridae containing up to three Phe/Trp residues per monomer."

... and (lines 238 - 244):

"... we propose that the binding affinities to the PM typically follow class I > class II \geq Class III. Exceptions to this trend exist as suggested by class III proteins with up to three Phe/Trp in their fusion loops, as discussed above, indicating that certain class III proteins reveal stronger membrane affinity compared to certain class II proteins "

We write now more explicitly in the captions of Table S6:

"Hydrophobic aromatic residues of fusion loops capable of inserting into the membrane are underlined (phenylalanine, F; tryptophan, W). Fusion loops of hantaviridae and most flaviviridae analyzed here contain one F and one W. Fusion loops of togaviridae contain two F and one W residue. Fusion loops of certain phenuiviridae (RVFV, HTRV, SFTSV) contain one F and one W residue, while in other phenuiviridae W is replaced by a hydrophobic leucine or isoleucine residue."

... and in the caption of Table S7:

"Hydrophobic aromatic residues of fusion loops inserting the membrane are underlined (phenylalanine, F; tryptophane, W). Among genera analyzed here, fusion loops of rhabdoviridae contain one to four F/W residues, baculoviridae and orthomyxoviridae only one F residue, whereas herpesviridae zero to three F/W residues."

6) Uncertainty estimates are lacking for the reported binding free energies throughout the manuscript.

REPLY: Uncertainties of binding free energies were computed by bootstrapping complete histograms from the set umbrella histograms, which provides a rather conservative error estimate (see Ref. 80). Uncertainties are shown in all PMFs as shaded areas (Fig. 2E, 4A, S1A, S10A) and as vertical bars in ΔG_{bind} values (Fig. 2D, 4B, S1B, S10B). They are described in the figure captions as follows, for instance for Fig. 2: "Vertical black bars in (C, D) and shaded areas in (E) denote ± 1 SE."

7) The PMFs shown in Figure 4A seem to have an entry barrier. This warrants some explanation.

Thank you for spotting this. We added (lines 311 - 318):

"Notably, these PMFs furthermore reveal a small entry barrier reflecting the energetic cost for moving the hydrophobic fusion loops across the polar headgroup region (Fig. 4A, distance ~ 1.3 nm). This barrier is reduced at higher cholesterol content, rationalized by the fact that cholesterol acts as spacer between phospholipids, reducing the head group packing density."

8) In the Discussion section, the authors argue that the strong membrane binding energy (115 kJ/mol) of a single trimer implies that three trimers are sufficient to overcome the energy required for stalk formation (ranging between 250-500 kJ/mol). I do not see how you can meaningfully compare these two different energies. For instance, a membrane spanning protein surely has an even much stronger binding energy to the membrane, but will likely fail to favour the formation of stalks.

Thank you for pointing this out, as the argument was indeed not precise enough. We fully agree that a strong membrane binding affinity of some protein does *not imply* any support in stalk formation, as exemplified by a transmembrane protein that the reviewer mentioned. However, our argument was meant to go the other way around: If the fusion protein binds with a weaker affinity compared to the cost of stalk formation, the fusion protein would, while it collapses and pulls on the membrane, first detach from the membrane before a stalk forms. Dissociating from the membrane during protein collapse would be energetically cheaper than stalk formation.

In other words, we agree: strong binding does not imply fusion. Instead, we mean: with weak binding, fusion would not be possible.

We now write more clearly (lines 586 - 596):

“Aeffner et al. estimated that the free energy required to form a stalk-like structure in the absence of proteins ranges between ~250 kJ/mol and ~500 kJ/mol, depending on the lipid composition (9). Thus, binding of at least three IAV HA2 trimers to the plasma membrane would be required to avoid that the membrane-bound HA2 trimers dissociate from the membrane during HA2 collapse, consistent with experimental findings showing that a minimum of three trimers is required to mediate fusion (55-57).”

9) Some essential details on how the modelling was performed are missing. For instance, how were disordered protein loops in Martini treated (as Martini protein models depend on assignment of secondary structural elements)? What was the force field used for cholesterol, and for the gangliosides (different models have been put forward by the Martini community, the choices should be discussed in light of the advantages/disadvantages of these models)?

We preserved the secondary and tertiary structure with an elastic network model, which is now described in the Methods (lines 730 - 734):

“elastic networks with a elastic bond force constant of 500 kJ/mol-nm², and elastic bond cut-offs set to 0.5 nm (lower) and 0.9 nm (upper). The linker to the fusion peptide of HA2 was kept flexible”.

For cholesterol, we used the reparametrized Martini model, which reproduced lipid phase behavior in mixed membranes (<https://doi.org/10.1063/1.4937783>). For gangliosides, we used a modified Martini parameterization designed to capture ganglioside clustering observed at the atomistic level (<https://doi.org/10.1021/acs.jpcc.6b07142>). We are aware modeling of cholesterol and gangliosides faces some challenges in the Martini model. However, considering that our results are in line with a wide range of experimental data. We now write (lines 561 - 574):

“We computed the free energy of membrane binding for three viral fusion proteins using coarse-grained simulations, and we validated the simulations against a wide body of experimental data: the simulations (i) identified the lipid binding site of RVFV Gc in line with crystallographic data (22), (ii) revealed increasing binding affinity with increasing cholesterol content (22,25), (iii) align with effects of the Dg61K mutant and PS lipids on RVFV Gc binding (22), and (iv) furthermore, binding affinities of the IAV HA2 fusion peptide are in good agreement with experimental data (47). Thus, the coarse-grained simulations likely provide a reasonable model for estimating binding affinities of viral fusion proteins.”

10) Moreover, apparently a beta version of the Martini 3 model was used - is there a link to find the actual parameters? A proper citation of the martinize tool is also lacking, this tool has been published in eLife already a while ago. Finally, elastic networks are mentioned but it is unclear what this entails.

Martini 3 beta force field files used in our simulations have been uploaded to Zenodo, with the corresponding link provided in the manuscript under *Data and Materials Availability*.

For protein topology generation, we used the *martinize.py* script, and the relevant citation (Ref. 67) has been correctly included. While *martinize2*—the successor to the original *martinize* tool—was published in *eLife*, we specifically used the original *martinize* tool and therefore did not cite the *eLife* paper (Kroon et al., *eLife* 2024).

Regarding the elastic network please, see our reply above.

Reviewers' comments:

Reviewer #3 (Remarks to the Author):

The manuscript has been nicely revised, all my comments have been adequately addressed. The only thing missing is proper citation of the cholesterol and ganglioside models used.

We have now added references to cholesterol and ganglioside models. We now write (lines 707 - 712):

For cholesterol, we used the reparametrized Martini model, which reproduced lipid phase behavior in mixed membranes (<https://doi.org/10.1063/1.4937783>). For gangliosides, we used a modified Martini parameterization designed to capture ganglioside clustering observed at the atomistic level (<https://doi.org/10.1021/acs.jpcc.6b07142>).